# A novel positive selection system for plant transformation based on microbial biuret hydrolase and biuret

Xiaowei Luo ◉*, Qiuyan Yu◉, Di Hu, Ming Gong, Zhurong Zou*

Engineering Research Center for Valorization of Unique Bio-Resources in Yunnan, Ministry of Education School of Life Sciences, Yunnan Normal University, Kunming, Yunnan Province, People's Republic of China

◉ These authors contributed equally to this work
* 2023120047@ynnu.edu.cn (XL); zzr09@ynnu.edu.cn (ZZ)

## Abstract

Conventional selectable marker genes (SMGs) in plant transformation, such as those conferring antibiotic resistance, raise biosafety concerns due to their permanent genomic integration. Alternative systems, such as the phosphite (Phi)/phosphite dehydrogenase (PtxD) system, have been developed to address these concerns. This system works by enabling transgenic plants to convert non-metabolizable, toxic Phi into usable orthophosphate (Pi), thereby providing both a detoxification mechanism and a positive growth advantage for selection. Building on this principle of detoxification-based positive selection, we have developed a novel system utilizing the microbial biuret hydrolase (BH) gene and its substrate, biuret (BU). Biuret is a phytotoxic by-product found in urea fertilizers. We introduced two genetic constructs into tobacco via Agrobacterium-mediated transformation: one expressing only the BH enzyme and another expressing a dual-function BH–AH fusion protein (BA2H, where AH is allophanate hydrolase). On nitrate-containing media, selection with 1 mM BU achieved a transformation efficiency of nearly 80% and shortened regeneration time by approximately 20 days compared to conventional Kanamycin selection (66.7% efficiency). The resulting transgenic lines demonstrated strong tolerance to BU during germination and seedling growth, even under nitrogen-deficient conditions. BU also acted as a selective herbicide, reducing weed biomass by ~70% in co-culture and field-simulation experiments while transgenic plants thrived. The BH/BU system provides an efficient, safe, and agriculturally relevant selection platform that avoids antibiotic resistance markers and leverages BU's dual role as a nitrogen source for crops and a growth inhibitor for weeds. This approach offers a sustainable alternative to conventional selection markers in plant biotechnology.

**Data availability statement:** All relevant data are within the paper and its Supporting Information files.

**Funding:** The author(s) received no specific funding for this work.

**Competing interests:** The authors have declared that no competing interests exist.

## Introduction

The development of transgenic crops plays a pivotal role in modern agriculture. A critical step in plant genetic transformation is the efficient selection of rare cells that stably integrate and express foreign genes. Selectable marker genes (SMGs) conferring resistance to antibiotics or herbicides remain the most widely used, largely owing to their well-established technical platforms [1]. However, these conventional SMGs persist in the plant genome after transformation, raising ongoing biosafety and public acceptance concerns [2,3]. Additionally, gene stacking—a key strategy for crop improvement—often requires multiple selectable markers, yet the number of highly efficient marker genes applicable to specific crops is limited, highlighting the need for new, versatile systems. Consequently, developing safer marker systems that can be excised or that utilize metabolically convertible compounds is a major research focus. An ideal positive selection system enables transgenic cells to detoxify a compound toxic to non-transformed cells while converting it into an essential nutrient, thereby providing a direct growth advantage exclusively to transgenic tissue. Currently, two main strategies address marker gene persistence: one is to remove the marker after transformation [4,5], and the other is to replace it with a marker that is inherently safe or possesses beneficial agronomic value [6–8]. This study follows the second strategy, aiming to develop a novel selection system based on a "detoxification-nutrition" paradigm targeting nitrogen (N), a macronutrient distinct from the well-studied phosphorus (P)-based systems. The ptxD/Phi system is a successful example of this principle [8]. Phosphite is toxic to plants because plants cannot utilize it as a phosphorus source, but the bacterial ptxD gene encodes a phosphite oxidoreductase that converts phosphite into metabolizable phosphate—enabling efficient selection, providing a phosphorus nutrient, and exhibiting herbicide potential.

In recent years, the Phi/PtxD system, utilizing ptxD as a dominant selectable marker gene, has achieved remarkable progress in multiple applications. For example, it addresses the issue of marker gene persistence in transgenic crops, expands the phosphorus utilization pathway in plants, reduces the over-reliance on Pi as the sole phosphorus source in current agricultural practices, and overcomes certain limitations of conventional herbicides. However, despite showing some selective efficacy in certain plant species such as soybean, the Phi/PtxD system exhibits relatively low efficiency and fails to clearly distinguish positive transformants from non-transformed tissues. Consequently, employing ptxD mutants with higher enzymatic activity to improve phosphite utilization efficiency in transgenic plants has emerged as a promising strategy to enhance the selection efficiency of the Phi/PtxD system. Additionally, the Phi/EcBAP [7] system has provided an alternative option for plant genetic engineering. However, when phosphite is supplied as the sole phosphorus source during selection, the endogenous BAP activity in individual transformed cells is insufficient to meet the cellular demand for metabolizable phosphate. Therefore, identifying BAP mutants with enhanced catalytic activity to improve phosphite utilization efficiency in EcBAP transgenic plants represents another important direction for future research.

Urea is the most widely used solid nitrogen(N) fertilizer globally. During its manufacture, a common by-product—BU—is formed and persists in commercial fertilizers at concentrations of 0.5–1.0%. This leads to the annual deposition of millions of kilograms of BU into agricultural soils. Unlike urea, BU is phytotoxic: it cannot be metabolized by plants and inhibits seed germination and seedling growth, causing significant yield losses in sensitive crops [9,10]. Interestingly, certain soil bacteria (e.g., *Herbaspirillum sp.*) possess a catabolic pathway to degrade BU. Its key enzyme, BH, catalyzes the hydrolysis of BU to allophanate [11], which is subsequently converted to ammonia, a readily assimilable N source for plants. This microbial capability presents a novel opportunity for plant biotechnology.

We hypothesized that expressing the bacterial BH gene in plants would confer two essential functions: detoxification of the phytotoxin BU, and utilization of the derived ammonia as a nitrogen source. This combination would enable the positive selection of transformants on BU-containing media. Furthermore, such a system promises distinct agronomic benefits: BU could serve as a slow-release nitrogen fertilizer for the transgenic crop [9,12] while simultaneously acting as a selective herbicide against non-transgenic weeds, which lack this detoxification capability. Therefore, the BH/BU-based system represents a promising safe and efficient selection platform with integrated weed management potential. Its foundation on a widely available agricultural by-product and its dual nutrient-herbicide function distinguish it from existing selection systems.

This study aimed to develop and validate this novel positive selection system for plant transformation. We constructed plant expression vectors for BH and a fusion gene (BA2H) designed for complete BU hydrolysis. Using tobacco as a model, we evaluated the system's transformation efficiency, conducted molecular characterization of transgenic lines, and assessed their BU tolerance and competitive growth advantage. Our results demonstrate that the BH/BU system is a viable, efficient, and safe alternative to traditional antibiotic- or herbicide-based selection markers.

## Materials and methods

### Gene synthesis and vector construction

The bacterial degradation of carbamoylurea proceeds via two enzymatic steps: biuret hydrolase (BH) converts biuret (BU) to allophanate, which is then hydrolyzed to ammonia and $CO_2$ by allophanate hydrolase (AH) (Fig 1). For complete hydrolysis of BU to ammonia in plants, we designed and synthesized a codon-optimized dual-function gene, designated *BA2H*. This gene encodes a fusion protein combining BH from *Herbaspirillum sp* [13] and AH from *Oleomonas sagaranensis* (GenBank BAD16655). The two enzyme domains are linked via a P2A [14] peptide sequence to ensure co-expression and post-translational cleavage, thereby maintaining their independent catalytic activities.

The synthetic BA2H fragment was cloned into the pBI121 vector between the *Sac*I and *Xba*I sites downstream of the CaMV 35S promoter, generating the plant expression vector pBI(BA2H). For comparison, the single-function BH gene was subcloned from this construct into pBI121, yielding pBI(BH). Both vectors were verified by Sanger sequencing (covering the entire coding region and promoter-terminator junctions) and restriction enzyme digestion analysis.

**Fig 1. Schematic of the biuret (BU) degradation pathway in bacteria and its proposed function in transgenic plants.** The bacterial pathway involves BH and AH. Expression of *BH* (or the fusion gene *BA2H*) in plants is hypothesized to confer BU tolerance by converting toxic BU into ammonia, a utilizable nitrogen source [11].

For recombinant protein expression in *E.coli*, the coding sequences of BA2H and BH were individually inserted into the pET28a(+) vector using *Nde*I and *Xho*I sites, resulting in pET(BA2H) and pET(BH), respectively. All constructs were confirmed by sequencing.

## Plant transformation and selection

**Determining the BU toxicity threshold for selection.** To establish an effective selection pressure for transformation, we first determined the phytotoxic threshold of BU on wild-type (WT) tobacco (*Nicotiana tabacum* cv. Xanthi) tissue. Seeds were surface-sterilized by immersion in 75% ethanol for 30 seconds and then in 10% $H_2O_2$ for 10 minutes, followed by five rinses with sterile water. The sterilized seeds were germinated on Murashige and Skoog (MS) medium. Leaf explants (0.7 × 0.7 cm) were excised from 4-week-old aseptic seedlings and placed on either standard MS medium or nitrogen-deficient media, each supplemented with a gradient of BU concentrations (0, 0.5, 1.0 and 2.0 mM). The explants were cultured at 25°C under a 16 h light/8 h dark photoperiod (light intensity: 1500 Lux). Callus formation and shoot regeneration were evaluated after 30 days.

The assay showed that WT regeneration was completely inhibited (0% regeneration rate) at BU concentrations ≥ 1 mM. Therefore, BU concentrations of 0.5, 0.8, and 1.0 mM were selected for subsequent transformation experiments to identify the optimal selection window.

**Plant transformation and selection regimes.** Leaf discs (0.7 cm × 0.7 cm) excised from 4-week-old aseptic tobacco seedlings were transformed using *Agrobacterium tumefaciens* strain GV3101. The discs were immersed in an *Agrobacterium* suspension ($OD_{600}$ = 0.4) for 5 minutes, blotted dry on sterile filter paper, and then co-cultivated on MS medium without selective agents for 3 days at 28°C in the dark. Following co-culture, explants were transferred to selection media and maintained at 26°C under a 16 h light/8 h dark photoperiod. All transformation experiments were conducted with three independent biological replicates, with three technical replicates (each replicate consisting of 20–25 explants) per treatment.The following control groups were included: (i) non-transformed wild-type (WT) explants cultured on selection media (with BU or kanamycin) as a negative control to confirm the efficacy of the selective agents; (ii) WT explants cultured on regeneration medium without any selective agent as a growth control (positive control) to monitor normal regeneration capacity; and (iii) explants transformed with the empty vector (or a construct containing the NPTII gene) and selected on kanamycin-containing medium as a positive control for the transformation procedure, enabling comparison of the novel BU-based system with the conventional selection method. Two selection regimes were compared:

(1) BU selection: Explants were cultured on standard MS medium supplemented with 0.5, 0.8, or 1.0 mM BU, and 200 mg/L timentin (Tim) to eliminate Agrobacterium.

(2) Control selection: Explants were cultured on standard MS medium containing 0.2 mM kanamycin and 200 mg/L Tim.

To assess whether hydrolyzed BU could serve as a nitrogen source for transgenic tissues, a parallel set of explants was cultured on media containing 0.5, 0.8, or 1.0 mM BU (supplemented with cefotaxime) under three different nitrogen regimes: (A) MS(−N1) medium, which lacks $NH_4NO_3$ but contains $KNO_3$; (B) nitrogen-free MS(−N0) medium; and (C) MS(−N2) medium, with $NH_4^+$ supplied as $(NH_4)_2SO_4$ as the sole nitrogen source.

**Calculation of transformation efficiency and positive confirmation rate:**

Transformation efficiency (%) = (Number of PCR-positive regenerated shoots/ Total number of explants inoculated) × 100

Positive confirmation rate (%) = (Number of PCR-positive shoots/ Total number of regenerated shoots) × 100

**Molecular characterization of transgenic plants.**

**Genomic DNA PCR.** Genomic DNA was extracted from leaves of putative transgenic and wild-type (WT) plants using the Tiangen Biotech Company's Reagent Kit. To confirm transgene integration and avoid false positives, multiplex PCR was performed using two primer sets simultaneously: one set specific to the *BH* or *BA2H* coding sequence, and another targeting the CaMV 35S promoter region on the vector backbone. Primer sequences and expected product sizes are listed in Supplementary S1 Table. The PCR protocol was: initial denaturation at 94°C for 3 min; 35 cycles of 94°C for 10 s, 54°C for 10 s, and 72°C for 45 s; final extension at 72°C for 5 min.

**Reverse transcription-PCR (RT-PCR).** Total RNA was isolated from roots, stems, and leaves of three independent T0 lines per construct using Tiangen TransScript® One Step gDNA Removal. After treatment with DNase I to remove genomic DNA, 1 µg of RNA was reverse-transcribed using the Tiangen cDNA Synthesis SuperMix kit. Semi-quantitative RT-PCR was performed with gene-specific primers (S4 Fig) and cycle numbers optimized to be within the linear amplification range (28 cycles for BH/BA2H, 25 cycles for the internal control, 18S rRNA). The PCR conditions were as described above.

**Phenotypic analyses.**

**Germination and seedling growth assays.** Seeds from wild-type (WT) and T1 transgenic lines (harvested from confirmed T0 plants) were surface-sterilized and sown on MS medium containing a range of BU concentrations (0, 0.5, 0.8, 1.0, or 2.0 mM) or 0.2 mM kanamycin as a control for comparison. The plates were incubated at 25°C under a 16-hour light/8-hour dark photoperiod with a light intensity of 1500 lux. To assess different growth parameters, half of the plates for each genotype and treatment were placed horizontally to evaluate germination rate and shoot development, while the other half were placed vertically to facilitate root length measurement.

After 14 days, the germination rate was calculated, and seedling growth was quantified by measuring root length, shoot height, and fresh weight per seedling. The experiment included three independent biological replicates, with ≥ 50 seeds per replicate per treatment.

**Weed suppression and competition experiments.**

(1) **In vitro weed sensitivity to BU:** Seeds of two common weed species, *Cynodon dactylon* (bermudagrass) and *Festuca arundinacea* (tall fescue), were surface-sterilized and placed on filter paper moistened with either distilled water (control) or distilled water supplemented with 1 mM or 2 mM BU. The plates were incubated at 25°C under a 16 h light/8 h dark cycle. Plant growth was assessed after 30 days.

(2) **Co-cultivation competition assay:** Seeds of WT tobacco, transgenic tobacco (BH or BA2H lines), C. dactylon, and F. arundinacea were combined in a 1:1:4:4 ratio (100 seeds total per pot) and sown in pots containing a sterilized soil mixture of sand, loam, and humus (1:2:1). Pots were irrigated every 3 days with 50 mL of either ddH$_2$O (control) or ddH$_2$O supplemented with 2 mM BU. Growth status was observed after 60 days.

(3) **Pot-based competition experiment under controlled conditions:** To simulate field competition under varying BU concentrations, a seed mixture containing tobacco (50 WT and 50 transgenic seeds) and weeds (200 seeds each of *C. dactylon* and *F. arundinacea*) was sown in large pots (40 cm diameter) filled with field soil. Pots were irrigated every 5 days with 200 mL of nitrogen-free nutrient solution (half-strength MS medium without nitrogen) supplemented with 0, 2, 4, 6, or 8 mM BU. Plants were grown at 25–30°C under natural light supplemented to maintain a 16 h photoperiod. Plant growth was assessed after 70 days. Each treatment included three independent biological replicates.

## Statistical analysis

All data were first tested for normality using the Shapiro–Wilk test and for homogeneity of variances using Levene's test. Data are presented as mean±standard deviation (SD). As the data met the assumptions of normality and homoscedasticity ($p > 0.05$), differences between groups were analyzed by one-way ANOVA followed by Tukey's post-hoc test for multiple

comparisons. A p-value < 0.05 was considered statistically significant. Each treatment group included at least 30 explants (for transformation experiments) or 50 seeds (for germination assays) to ensure sufficient statistical power.

## Results

In this study, we demonstrate that the biuret (BU)-based selection system serves as an effective alternative to conventional antibiotic resistance markers for plant transformation.This system functions by detoxifying BU—a phytotoxic by-product of urea fertilizers—into a metabolically usable nitrogen source through the expression of a bacterial biuret hydrolase (BH) gene in transgenic plants.

### Enhanced transformation efficiency

To establish an effective selection pressure for transformation, we first determined the phytotoxic threshold of BU on wild-type (WT) tobacco (Nicotiana tabacum cv. Xanthi) tissue (Fig 2, S2). Observations of callus formation and shoot

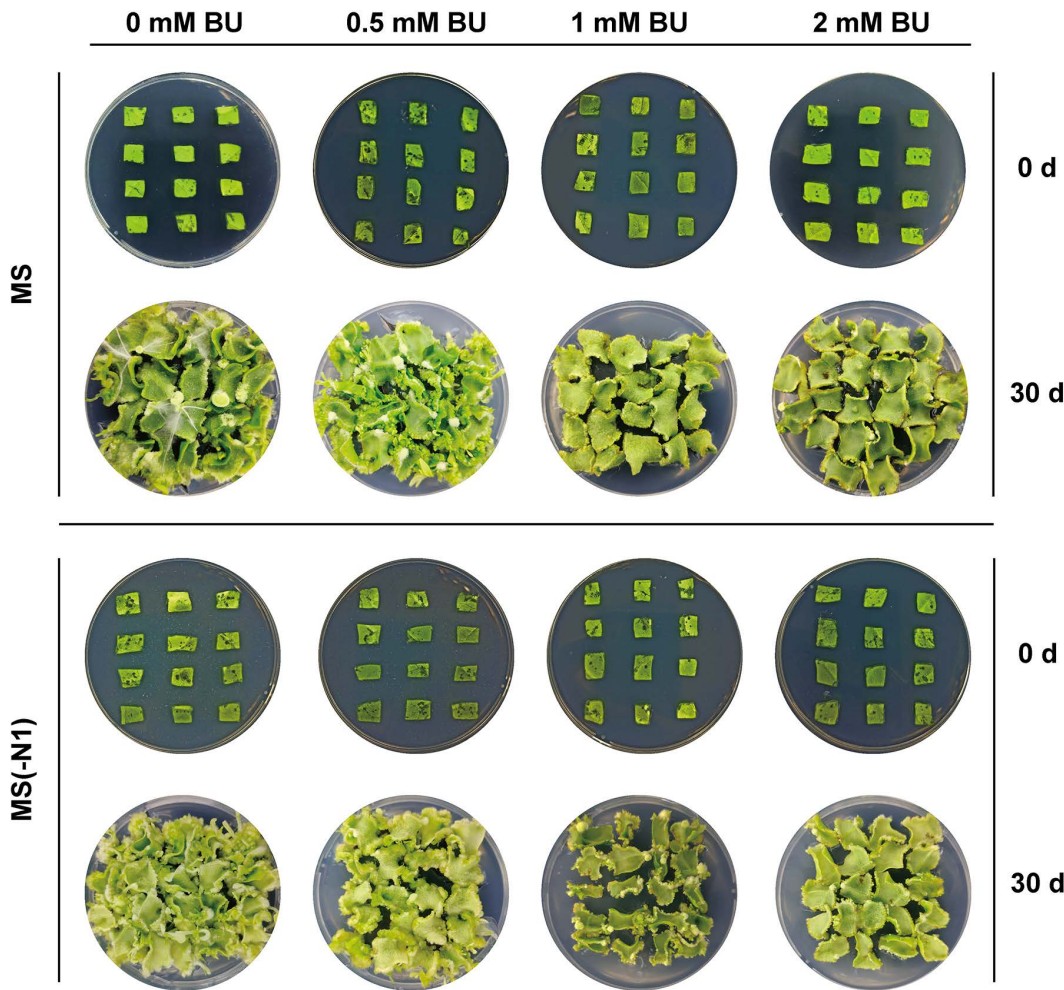

**Fig 2. Effect of nitrogen source composition on the sensitivity of wild-type tobacco leaf explants to BU.** Explants were cultured for 30 days on media with varying nitrogen compositions: standard MS and MS(-N1), each supplemented with a gradient of BU concentrations (0 to 1 mM).

regeneration after 30 days of culture showed that on standard MS medium, WT leaf explants exhibited severely inhibited callus development under BU stress at 1 mM or higher concentrations. In contrast, explants treated with 0.5 mM BU maintained comparable callus differentiation and shoot regeneration to the BU-free control. A similar response pattern was observed on nitrogen-deficient MS(-N1) medium, though both callus initiation and differentiation proceeded more slowly compared to the standard MS medium. On the other two nitrogen-deficient media, MS(-N0) and MS(-N2), no callus formation occurred, and the leaf explants exhibited chlorosis and necrosis. Based on these findings, we selected 0.5, 0.8, and 1 mM BU as the selection pressures for subsequent tobacco transformation experiments on MS, MS(-N1), MS(-N0) and MS(-N2) media.

In tobacco transformation experiments, BH- and BA2H-transgenic plants were selected on both standard MS and MS(-N1) media supplemented with 0.5, 0.8, or 1.0 mM BU, using standard 0.2 mM kanamycin selection as a control. The highest transformation efficiencies were achieved on MS medium: 52.6±4.2% for BH with 0.8 mM BU, and 78.5±5.1% for BA2H with 1.0 mM BU (Table 1). The efficiency for BA2H under 1.0 mM BU selection was significantly higher than that obtained with the 0.2 mM kanamycin control (66.7±6.3%) ($p < 0.05$), representing a 12.2 percentage point increase (18.3% relative improvement) over the conventional selection method. A complete summary of all tested concentrations and constructs, including suboptimal conditions, is provided in S2 Table. Moreover, the regeneration process under BU selection was notably faster across all effective concentrations, requiring only 62–70 days from explant to rooted shoot, compared to 85–90 days for the kanamycin system—a reduction of approximately 20 days (Table 1, S2 Table, Fig 3). This accelerated regeneration is likely associated with the dual function of BU in this system: it acts as a phytotoxic agent to suppress non-transformed cells, and its hydrolysis product, ammonia, may contribute to nitrogen availability in transgenic tissues.

## Nitrogen background is critical for selection efficacy

The efficacy of the BH/BU selection system was critically dependent on the nitrogen composition of the culture medium. While transgenic shoot regeneration efficiency reached 75–80% on nitrate-containing media (standard MS and MS(−N1); consistent with Table 1), it dropped to 0% on either nitrogen-free (MS(−N0)) or ammonium-only (MS(−N2)) media (S2 Fig). This requirement for nitrate aligns with established plant physiology, where $NO_3^-$ plays a crucial role in mitigating $NH_4^+$ toxicity by maintaining rhizosphere pH and facilitating ammonium assimilation [15]. The complete failure of regeneration on ammonium-only medium (MS(−N2)) is notable. One possible explanation is that the combination of BU-derived ammonia and the exogenous $NH_4^+$ supply may lead to ammonium accumulation, which could inhibit cell growth and regeneration [16]. However, further experiments—such as measuring intracellular ammonium concentrations or tracking nitrogen

**Table 1. Comparison of maximum transformation efficiency and regeneration time between the BU-based and Kanamycin-based selection systems.**

| Selection System | BU Concentration (mM)/ Kan (mM) | Transformation Efficiency (%) | Regeneration Time (days) |
|---|---|---|---|
| BH / BU | 0.8 mM BU | 52.6±4.2[c] | 62 - 70 |
| BA2H / BU | 1.0 mM BU | 78.5±5.1[a] | 62 - 70 |
| NPTII / Kanamycin | 0.2 mM Kan | 66.7±6.3[b] | 85 - 90 |
| Wild-type negative control | 1.0 mM BU | 0.0±0.0[d] | – |

*Values are presented as mean±standard deviation (SD) from three independent experiments, each with four technical replicates (20–25 explants per replicate). Different lowercase letters within the Transformation Efficiency column indicate statistically significant differences among treatments at $p < 0.05$, as determined by one-way ANOVA followed by Tukey's post-hoc test. The wild-type negative control (non-transformed explants on 1.0 mM BU) showed no regeneration, confirming the efficacy of the selective agent. The wild-type growth control (non-transformed explants on medium without selection) showed normal regeneration, confirming explant viability. All statistical analyses were performed using SPSS software.*

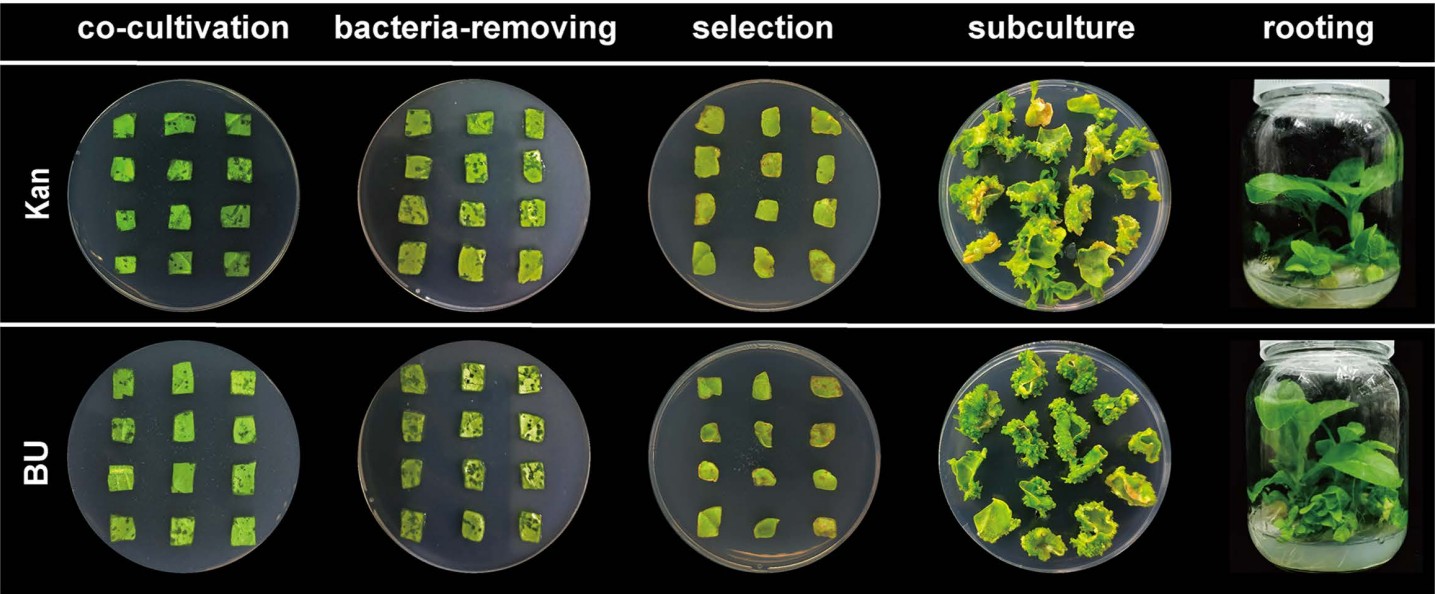

**Fig 3. Regeneration dynamics of of BA2H-transgenic tobacco (Nicotiana tabacum cv.** Xanthi) leaf explants under different selection regimes. Following Agrobacterium-mediated transformation, explants were cultured sequentially on MS media with either 1 mM BU or 0.2 mM Kanamycin.

metabolism—would be necessary to confirm this mechanism. It is also possible that other factors, such as pH changes or metabolic imbalances, contribute to the observed phenotype.

Importantly, the system's dependence on a nitrate background is agronomically compatible. Nitrate is the predominant form of available nitrogen in most agricultural soils and is a standard component of plant tissue culture media. This inherent compatibility ensures that the BH/BU system can be integrated into existing transformation and cultivation protocols without necessitating major modifications to nitrogen management practices.

## BU tolerance, herbicide potential, and construct-dependent performance

Transgenic tobacco lines expressing *BH* or *BA2H* exhibited strong tolerance to BU during germination and seedling growth across various nitrogen regimes(Figs 4–7). Notably, under nitrogen-limited conditions [MS(−N1) medium with 2 mM BU], transgenic plants maintained vigorous growth (germination rate >80%, root length $1.5 \pm 0.4$ cm, shoot height $1.2 \pm 0.3$ cm), in stark contrast to severely inhibited wild-type (WT) plants (germination <30%, root length $0.4 \pm 0.2$ cm, shoot height $0.2 \pm 0.1$ cm; $p < 0.001$). This demonstrates the system's efficacy in converting a phytotoxin into a utilizable nutrient under nitrogen stress.

BU also displayed potent, selective herbicidal activity. In vitro, 2 mM BU reduced the germination rate and seedling fresh weight of *Cynodon dactylon* and *Festuca arundinacea* by 70–85% compared to controls(Fig 8A). In co-culture and pot-based competition experiments, irrigation with 2–8 mM BU suppressed total weed biomass by 60–85%, while transgenic tobacco biomass increased by 2.3-fold relative to WT, or showed 15–30% enhancement in growth parameters. These results validate BU's dual utility as a selective agent and a slow-release nitrogen fertilizer with integrated weed management potential(Fig 8B,C).

Interestingly, at higher BU concentrations (≥ 4 mM), a clear phenotypic divergence emerged: BH-expressing lines grew normally, whereas BA2H lines exhibited chlorosis and growth inhibition. We speculate that the more rapid and

   

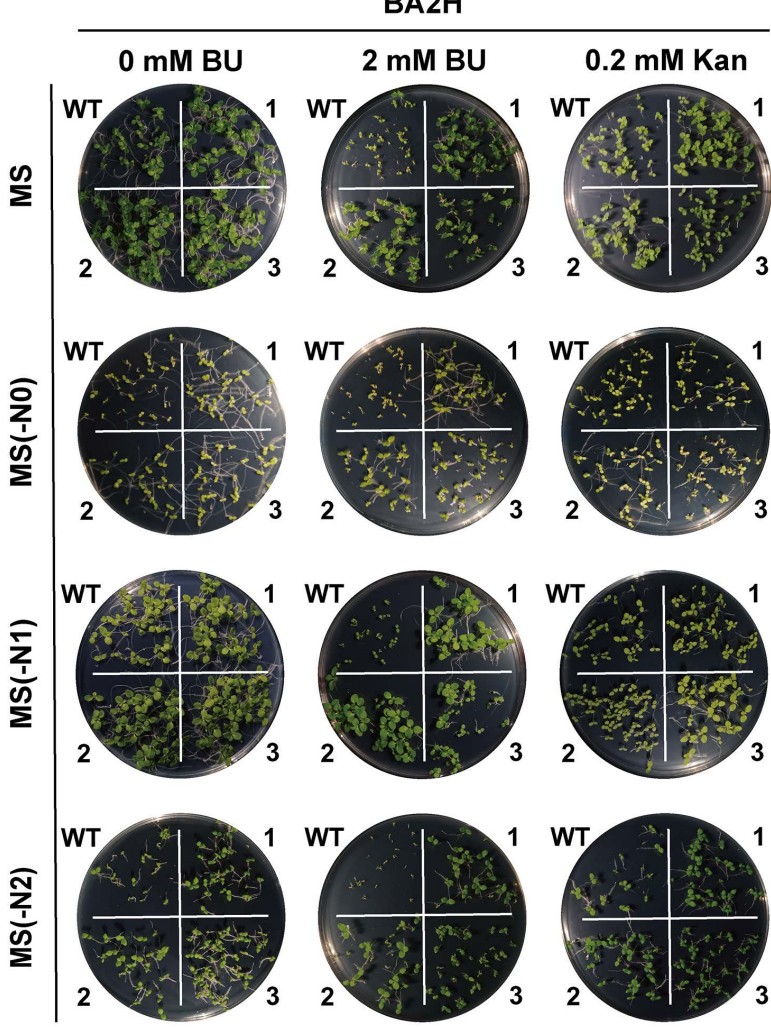

**Fig 4. Germination of *BA2H*-transgenic tobacco seeds on horizontal plates under biuret (BU) selection.** Seeds of wild-type (WT) and *BA2H*-transgenic T1 lines were germinated for 14 days on standard MS, nitrogen-deficient [MS(−N1)], nitrogen-free [MS(−N0)], or ammonium-only [MS(−N2)] media supplemented with 0.2 mM Kan or 2 mM BU. Images 1, 2, 3 represent three independent transgenic lines per construct..

complete hydrolysis of BU by the dual-function BA2H enzyme leads to localized ammonia accumulation that exceeds the plant's assimilation capacity, resulting in ammonium toxicity. However, this proposed mechanism remains hypothetical and requires direct experimental validation. Future studies measuring intracellular ammonium concentrations, assessing the activity of ammonia-assimilating enzymes (e.g., glutamine synthetase), or tracking nitrogen metabolism pathways in BA2H-transgenic plants under BU selection would be necessary to confirm whether ammonium toxicity is indeed the underlying cause.

If this mechanism is substantiated, it would suggest that the choice of transgene (BH vs. BA2H) could be optimized based on the target BU application level—BA2H for low-to-moderate (1–2 mM) and BH for high (≥ 4 mM) BU scenarios in weed-prone fields. Such optimization would need to be empirically tested in future greenhouse or field trials to balance effective selection with potential phytotoxicity.

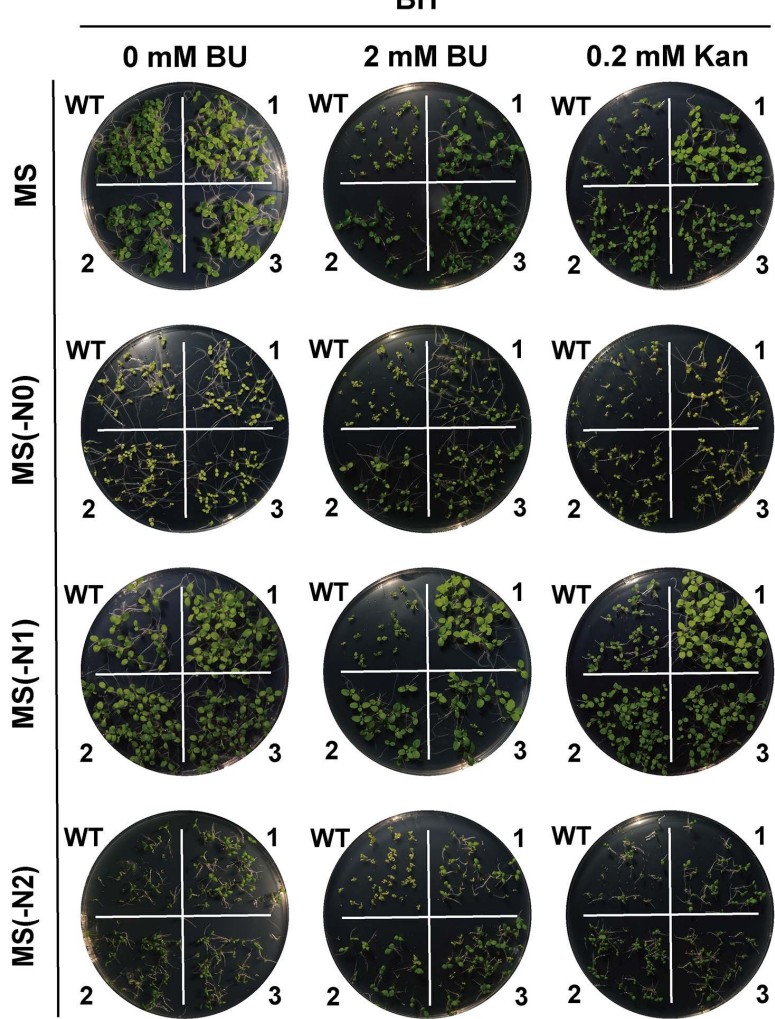

**Fig 5. Germination of *BH*-transgenic tobacco seeds on horizontal plates under biuret (BU) selection.** Seeds of wild-type (WT) and *BH*-transgenic T1 lines were germinated for 14 days on standard MS, nitrogen-deficient [MS(−N1)], nitrogen-free [MS(−N0)], or ammonium-only [MS(−N2)] media supplemented with 0.2 mM Kan or 2 mM BU. Images 1, 2, 3 represent three independent transgenic lines per construct.

## Discussion

We have developed and validated, for the first time, a novel positive selection system for plant transformation based on the microbial BH gene and its substrate BU, a common phytotoxic by-product of urea manufacturing. The BH/BU system enables efficient recovery of transgenic plants, achieving a transformation efficiency of approximately 80%—significantly higher than that of the conventional Kan system (66.7%)—and shortens the regeneration timeline by about 20 days, all while eliminating the need for antibiotic or herbicide resistance markers. As highlighted in a recent comprehensive review [17], conventional selection based on antibiotic or herbicide resistance genes, while effective, often involves labor-intensive screening processes and raises biosafety concerns. Commonly used agents such as kanamycin, hygromycin, and herbicides (e.g., bar, epsps) require careful optimization to avoid negative effects on plant regeneration [17]. The BH/BU system addresses these limitations by replacing antibiotic/herbicide resistance with a metabolically convertible marker that utilizes an agricultural by-product, thereby simplifying the screening process and enhancing biosafety.

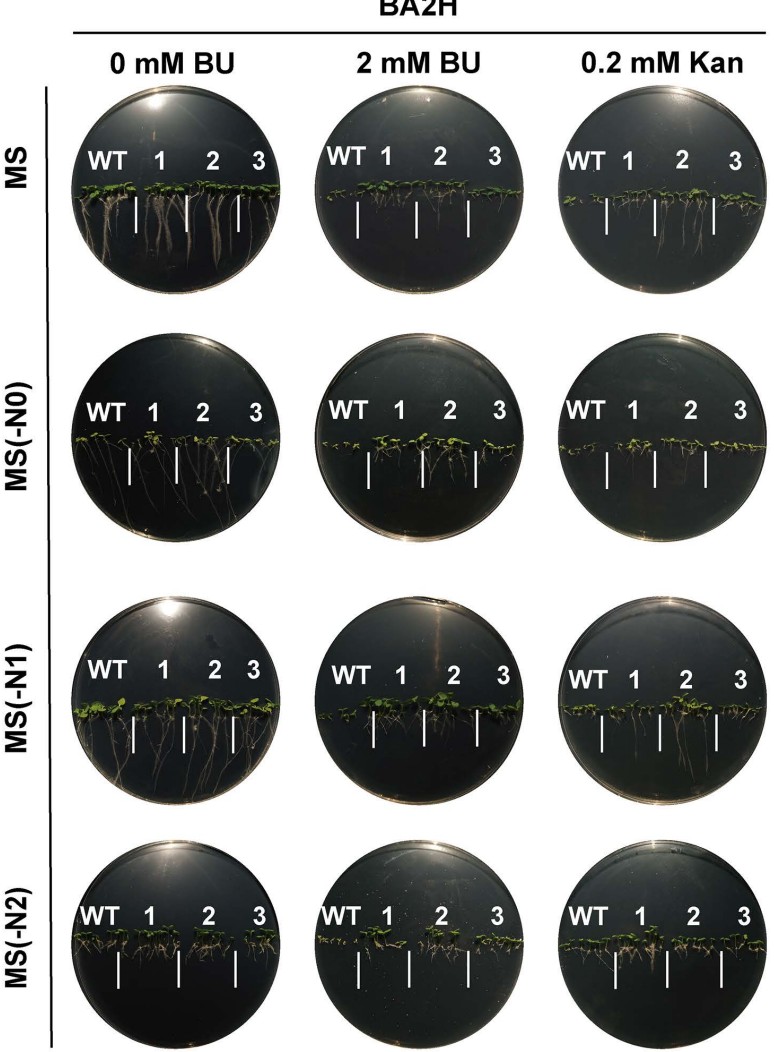

**Fig 6. Root growth of *BA2H*-transgenic tobacco seeds on vertical plates under biuret (BU) selection.** Seeds of wild-type (WT) and *BA2H*-transgenic T1 lines were germinated for 14 days on standard MS, nitrogen-deficient [MS(−N1)], nitrogen-free [MS(−N0)], or ammonium-only [MS(−N2)] media supplemented with 0.2 mM Kan or 2 mM BU. Images 1, 2, 3 represent three independent transgenic lines per construct..

This system innovatively repurposes an agricultural waste product into a multifunctional agronomic tool. BU serves concurrently as: (i) a selective agent in tissue culture that inhibits non-transformed cells, (ii) a slow-release nitrogen source for transgenic crops via BH-mediated hydrolysis to ammonia, and (iii) a selective herbicide against non-transgenic weeds, which lack the detoxification capability. Compared to other "detoxification-nutrition" systems such as ptxD/Phi, the BH/BU system offers distinct advantages: it utilizes a widely available, low-cost urea by-product and provides inherent weed-suppression activity, potentially reducing dependence on external herbicides. In contrast to sugar-based selection systems (e.g., pmi/mannose or xylA/xylose) which also avoid antibiotic resistance but require specific carbohydrate metabolism pathways [17], the BH/BU system targets nitrogen metabolism—a fundamental macronutrient pathway conserved across plants—offering broader applicability.

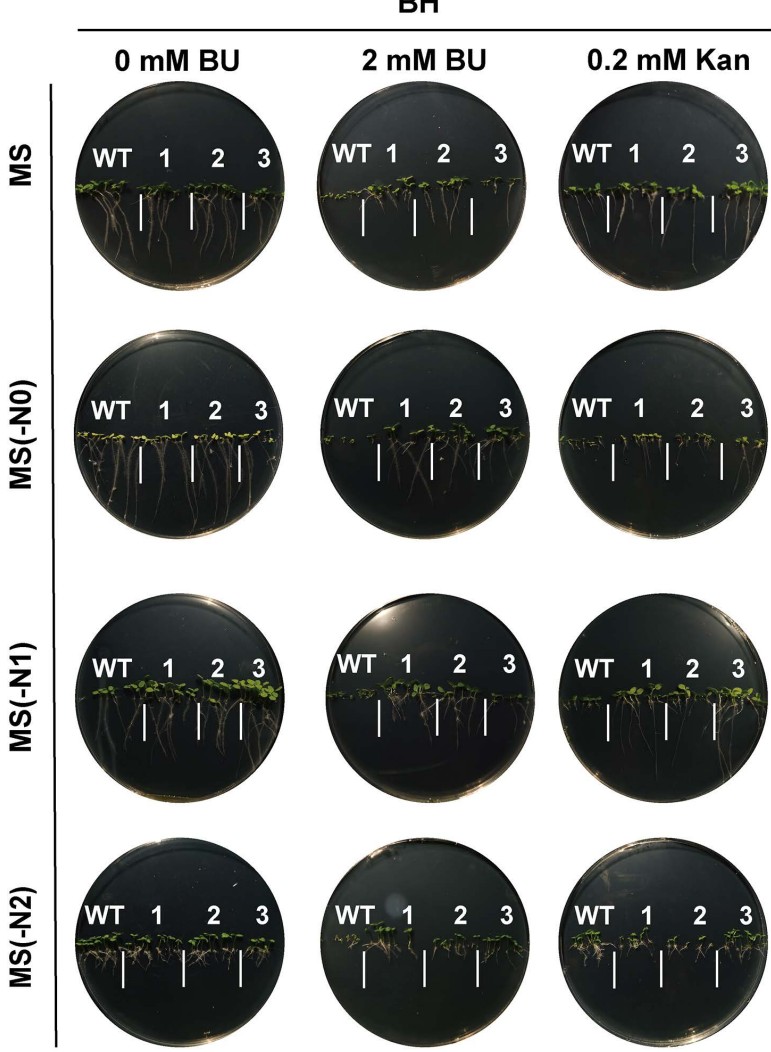

**Fig 7. Root growth of *BH*-transgenic tobacco seeds on vertical plates under biuret (BU) selection.** Seeds of wild-type (WT) and *BH*-transgenic T1 lines were germinated for 14 days on standard MS, nitrogen-deficient [MS(−N1)], nitrogen-free [MS(−N0)], or ammonium-only [MS(−N2)] media supplemented with 0.2 mM Kan or 2 mM BU. Images 1, 2, 3 represent three independent transgenic lines per construct.

The recently developed RUBY-MAT system, which integrates a betalain-based RUBY reporter with morphogenic regulators, enables non-invasive real-time visualization of transgenic events in maize as early as 7 days after transformation [18]. This approach shares conceptual similarities with our BH/BU system: both aim to simplify screening by reducing reliance on antibiotics and complex detection procedures. However, while RUBY-MAT provides visual confirmation without conferring a growth advantage, the BH/BU system actively promotes transgenic cell proliferation through nitrogen release. These systems are therefore complementary—integrating metabolic selection with visual reporters could create powerful dual-function platforms that further streamline plant transformation.

Notably, the transformation efficiencies observed in this study exhibited considerable variability, particularly at lower BU concentrations (e.g., BH 0.5 mM: 2.2±2.6%; BA2H 0.5 mM: 8.7±4.1%; BA2H 0.8 mM: 13.6±6.4%) (S2 Table). This

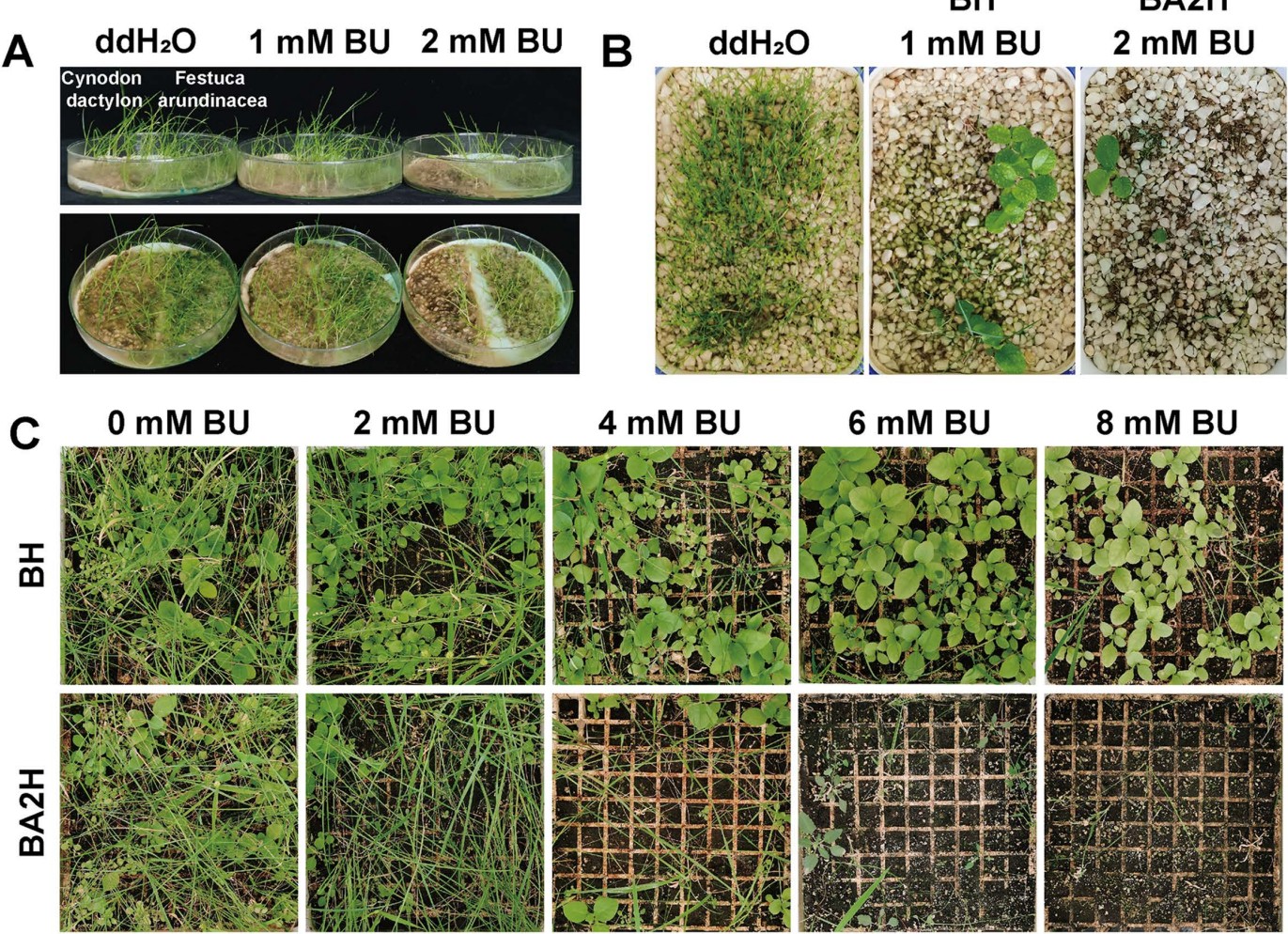

**Fig 8. Herbicidal activity of BU and competitive growth advantage of BH-expressing tobacco under BU application.** (A) In vitro sensitivity of weed species to BU. Seeds of Cynodon dactylon and Festuca arundinacea were germinated on filter paper moistened with 0, 1, or 2 mM BU for 30 days. (B) Co-culture competition assay. A mixed population of WT tobacco, transgenic tobacco (BH,BA2H), and weeds was grown in soil irrigated with 0 or 2 mM BU for 60 days. (C) Pot-based competition under varying BU concentrations. Weed biomass and tobacco growth were evaluated after 70 days in a mixed planting system irrigated with a nitrogen-free solution containing 0 to 8 mM BU.

variation likely reflects the stochastic nature of T-DNA integration and transgene expression, which are influenced by factors such as insertion site, copy number, and epigenetic effects. The high standard deviations at suboptimal BU concentrations suggest that under weaker selection pressure, the growth advantage conferred by BU hydrolysis is insufficient to reliably support all transgenic events, leading to inconsistent regeneration outcomes. In contrast, at optimal concentrations (BA2H 1.0 mM, 78.5±5.1%; BH 0.8 mM, 52.6±4.2%), the stronger selection pressure minimized this variability by more effectively suppressing non-transformed cells and promoting uniform transgenic tissue growth. Similar patterns of efficiency variability have been reported in other metabolic selection systems, such as the ptxD/Phi system, where transformation success depends critically on the balance between selective agent concentration and enzyme activity. These observations underscore the importance of optimizing selection conditions for each construct to achieve consistent and reproducible transformation outcomes.

In summary, this work provides a safe, efficient, and agronomically integrated alternative to conventional selection markers. By leveraging nitrogen metabolism, it contributes a complementary platform to the growing toolkit of sustainable plant biotechnologies. The system's compatibility with standard nitrate-based media and its relevance to mainstream agricultural soils further support its potential for broad application in crop genetic improvement programs.

To translate this proof-of-concept into a robust agronomic tool, future studies should focus on: (i) evaluating the long-term effects of repeated BU application on soil microbial communities and nutrient cycling; (ii) using protein engineering to optimize BH/BA2H enzyme properties and reduce potential ammonia toxicity; and (iii) integrating the system with site-specific recombination to generate marker-free plants.

## Supporting information

**S1 Fig. Effect of nitrogen source composition on the sensitivity of wild-type tobacco leaf explants to BU.** Explants were cultured for 30 days on media with varying nitrogen compositions: standard MS(-N0) and MS(-N2), each supplemented with a gradient of BU concentrations (0–1 mM).
(TIF)

**S2 Fig. Regeneration dynamics of tobacco (Nicotiana tabacum cv. Xanthi) leaf explants under different selection regimes.** Following Agrobacterium-mediated transformation, explants were cultured sequentially on MS(-N0) media,MS(-N1) media and MS(-N2) media with either 1 mM BU or 0.2 mM Kanamycin.
(TIF)

**S3 Fig. Regeneration dynamics of BH-transgenic tobacco (Nicotiana tabacum cv. Xanthi) leaf explants under different selection regimes.** Following Agrobacterium-mediated transformation, explants were cultured sequentially on MS media with either 1 mM BU or 0.2 mM Kanamycin.
(TIF)

**S4 Fig. RT-PCR analysis of BH and BA2H expression in root, stem, and leaf tissues of BH and BA2H transgenic tobacco.** (A) Extracted total RNA; (B) Primer pairs Nt18S-iFw/ Nt18S-iRv were used for RT-PCR of the internal reference gene 18S rRNA to obtain the correct product (552 bp); (C) BA2H transgenic tobacco was amplified using specific primers BH-iFw/ BH-iRv, AH-iFw/ AH-iRv for the BH and AH genes, respectively (AH gene size 373 bp).(D)BH transgenic tobacco amplification using specific primers BH-iFw/ BH-iRv and electrophoretic detection (BH gene size 477 bp). R:root; S:stem; L:leaf; arrows indicate the target PCR bands.
(TIF)

**S5 Fig. Molecular identification of positive transgenic plants under BU-based selection.** Regenerants obtained under different selection regimes were screened by PCR using three primer sets specific to the transgene. Each lane represents an independent regenerated plantlet. Plants showing amplification with all three primer pairs were considered transgenic-positive. (A) Control: Positive BA2H transgenic plants selected on 0.2 mM kanamycin. (B) BA2H transgenic plants selected on 1 mM BU. (C) BH transgenic plants selected on 0.8 mM BU.
(TIF)

**S1 Table. Primers used in this study.**
(DOCX)

**S2 Table. Comparison of transformation efficiency and regeneration time between the BU-based and kanamycin-based selection systems.**
(DOCX)

**S3 File. The original gel images.**
(DOCX)

## Author contributions

**Conceptualization:** Zhurong Zou.

**Data curation:** Qiuyan Yu, Zhurong Zou.

**Formal analysis:** Qiuyan Yu.

**Funding acquisition:** Zhurong Zou, Ming Gong.

**Investigation:** XiaoWei Luo, Qiuyan Yu, Di Hu.

**Methodology:** XiaoWei Luo.

**Project administration:** XiaoWei Luo.

**Resources:** XiaoWei Luo.

**Software:** XiaoWei Luo.

**Supervision:** XiaoWei Luo, Qiuyan Yu, Ming Gong.

**Validation:** XiaoWei Luo, Qiuyan Yu, Di Hu.

**Visualization:** XiaoWei Luo.

**Writing – original draft:** XiaoWei Luo.

**Writing – review & editing:** XiaoWei Luo, Zhurong Zou.

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
