## [Decision Letter · Decision Letter 0]

10 Mar 2026

PONE-D-26-07660A Novel Positive Selection System for Plant Transformation Based on Microbial Biuret Hydrolase and Biuret

PLOS ONE

Dear Dr. Luo,

Thank you for submitting your manuscript to PLOS ONE. After careful consideration, we feel that it has merit but does not fully meet PLOS ONE’s publication criteria as it currently stands. Therefore, we invite you to submit a revised version of the manuscript that addresses the points raised during the review process.

We look forward to receiving your revised manuscript.

Kind regards,

Faiz Ahmad Joyia, Ph.D.

Academic Editor

PLOS One

Journal Requirements:

3. Please amend the manuscript submission data (via Edit Submission) to include author Dr. Di Hu and Dr. Ming Gong.

4. Please upload a copy of Figure 6, to which you refer in your text on page 18 in PDF submission. If the figure is no longer to be included as part of the submission please remove all reference to it within the text.

5. We note that Figure 2, 3, 4 and 5 in your submission contain copyrighted images. All PLOS content is published under the Creative Commons Attribution License (CC BY 4.0), which means that the manuscript, images, and Supporting Information files will be freely available online, and any third party is permitted to access, download, copy, distribute, and use these materials in any way, even commercially, with proper attribution. For more information, see our copyright guidelines: http://journals.plos.org/plosone/s/licenses-and-copyright.

1. You may seek permission from the original copyright holder of Figure 2, 3, 4 and 5 to publish the content specifically under the CC BY 4.0 license.

**Additional Editor Comments:**

1. Various figures in manuscript of low quality and need substantial improvements for publication.

2. Experiment layout, sample size, treatment plan, control treatments and replications are not clearly elaborated.

3. Use of low concentration of (25 ug/mL) of only Rifampicin makes the study questionable.

4. Statistical analyses are not adequate.

5. Claims are not justified with the presented data.

Reviewers' comments:

Reviewer's Responses to Questions

**Comments to the Author**

1. Is the manuscript technically sound, and do the data support the conclusions?

Reviewer #1: Yes

Reviewer #2: Yes

2. Has the statistical analysis been performed appropriately and rigorously? 

Reviewer #1: No

Reviewer #2: Yes

3. Have the authors made all data underlying the findings in their manuscript fully available?

Reviewer #1: No

Reviewer #2: Yes

4. Is the manuscript presented in an intelligible fashion and written in standard English?

Reviewer #1: Yes

Reviewer #2: No

5. Review Comments to the Author

Reviewer #1: Proper selectable marker genes play a critical role during plant bioengineering to ensure only the transformed cells proliferate and regenerate into whole plants. Most plant transformation protocols use either herbicide or antibiotic resistance genes as selectable markers. This paper evaluated the use of bacterial biuret hydrolase (BH) and its substrate biuret-a phytotoxic by-product of urea fertilizer metabolism-as a novel selection system for plant transformation. During in vitro assay, bacterial BH and its fusion with another enzyme BH-AH (allophanate hydrolase), hydrolyzed BU releasing ammonia. When used for tobacco leaf disc transformation using Agrobacterium, 1 mM BU showed slightly increased transformation efficiency (~80%) compared to kanamycin selection (~67%), while reducing the time from 85-90 days to 62-70 days. The authors also tested the potential of BU as a herbicide through a competition assay using transgenic tobacco and two weed species, Cynodon dactylon and Festuca arundinacea. BH transgenic tobacco plants demonstrated enhanced competitiveness in the presence of >2 mM BU, whereas transgenic BA2H tobacco plants exhibited sensitivity at >2 mM BU, suggesting that potential over-accumulation of ammonia by AH can cause phytotoxicity to the transgenic plants.

Overall, this paper provides some important data to utilize bacterial BH as a new positive selectable marker gene for plant transformation. However, data presentation and statistical analysis need to be improved.

Below are my specific comments.

1. Figure 3. Remove T1-T5 from the labels, as they are confusing. T0, T1, or T2 are used for transgenic lines. Showing the days after Agrobacterium infection would be sufficient. Alternatively, change “T1 0 d” to “Co-cultivation”; “T2 2 d” to “Selection 1”; “T3 5 d” to “Selection 2”; “T4 20 d” to “Selection 3”; “T4 35 d” to “Regeneration”; “T5 85 d” to “Rooting”.

2. Figure 4 needs to be improved. Some images are too small and it’s not easy to see the plants. Please clearly indicate which plants are WT and transgenic plants. In Figure 4B, change “CK” inside of a plate to “WT”.

3. Table 1. How many independent experiments were performed? Was this from a single transformation experiment or multiple independent experiments? It’s not clearly stated in the Materials and Methods section. Please include more details: how many explants per infection experiment and how many independent experiments were performed. There is no indication which treatment has significantly higher transformation efficiency. In the text, it says that BA2H has a significantly higher transformation efficiency than did Kanamycin selection (p<0.01). Table 1 footnote says that significance was determined by Tukey’s test with two p-values: p<0.01, *p<0.001. Which p-value was used for comparison?

4. Add alphabets (e.g., “a”, “b”, or “c”) next to “Transformation Efficiency” to indicate which treatment groups had statistically significant differences.

5. Table 1. Please include transformation efficiency for all tested treatments, not just the highest transformation efficiency for each treatment, because this can provide useful information for other researchers who want to test BU/BH for transformation.

6. In the section “Plant transformation and selection regimes”, the use of 25 ug/mL of Rifampicin seems wrong because Agrobacterium strain GV3101 is resistant to rifampicin. Typically, 10-50 ug/mL of rifampicin and 50 ug/mL gentamicin are added to grow GV3101. Carbenicillin, Timentin, or Cefotaxime are commonly used to eliminate Agrobacterium strains after co-cultivation. Have you checked the presence of Agrobacterium from the transgenic plants?

Reviewer #2: The study addresses a clearly defined and relevant research problem, providing new evidence that fills an important gap in the existing literature. It uses appropriate and modern approaches to generate findings with practical implications for policy and future research. The topic is timely because it aligns with current priorities and responds to emerging challenges in the field.

1. The experimental design is not described with sufficient clarity to allow replication. For example, the manuscript states that experiments were performed “in triplicate,” but it is unclear whether these are biological replicates (independent experiments) or technical replicates (multiple measurements of the same sample). This distinction directly affects statistical validity.

2. Sample size justification is missing. There is no explanation of how the number of samples or experimental units was determined. For instance, no power analysis or rationale is provided to demonstrate that the study is adequately powered to detect the reported differences.

3. Control treatments are inadequately defined. In some sections, a “control group” is mentioned without clearly specifying its conditions (e.g., untreated, vehicle-treated, or baseline). This makes interpretation of treatment effects ambiguous.

4. Statistical methods are insufficiently justified. The manuscript reports the use of parametric tests (e.g., ANOVA or t-test), but there is no indication that assumptions such as normality or homogeneity of variance were tested. Without this, the validity of the statistical conclusions is uncertain.

5. Exact p-values are not consistently reported. For example, results are described as “significant at p < 0.05” without providing the exact p-value or confidence intervals. This limits transparency and interpretability.

6. Effect sizes are not provided. The manuscript focuses solely on statistical significance without quantifying the magnitude of differences. For instance, reporting a significant increase without indicating percentage change or standardized effect size weakens the biological interpretation.

7. Some conclusions appear to overextend beyond the presented data. For example, the discussion implies a mechanistic explanation for observed results, although no mechanistic experiments were conducted to directly support that claim.

8. Figures lack sufficient clarity. Some graphs do not clearly label units on axes, error bars are not defined (SD vs SE), and figure legends do not adequately explain sample size or statistical comparisons.

9. The data availability statement is absent or incomplete. There is no clear indication of whether raw data, supplementary datasets, or underlying numerical values are accessible in a repository, which is required for transparency.

10. The introduction contains general background information but does not clearly define a specific knowledge gap. For example, while the topic is broadly introduced, the manuscript does not explicitly state what unanswered question this study addresses.

11. The discussion does not critically compare findings with recent literature. Some relevant studies from the past five years are not cited, and where comparisons are made, they are descriptive rather than analytical.

12. Terminology is occasionally imprecise. For example, causal language such as “this treatment induces…” is used even though the study design demonstrates association rather than mechanistic causation.

13. Methodological parameters are incomplete. Critical experimental details such as incubation times, reagent concentrations, environmental conditions, or instrument settings are either briefly mentioned or omitted, which compromises reproducibility.

14. There is limited explanation of variability. In cases where high variance is visible in figures, the manuscript does not discuss possible biological or technical reasons for this dispersion.

15. Language and phrasing occasionally reduce clarity. Some sentences are grammatically awkward or overly long, which makes interpretation of technical details difficult.

6. PLOS authors have the option to publish the peer review history of their article (what does this mean?). If published, this will include your full peer review and any attached files.

Reviewer #1: No

Reviewer #2: No

---

## [Author Response · Author response to Decision Letter 1]

23 Mar 2026

Dear Editor and Reviewers,

We sincerely thank you for taking the time and effort to review our manuscript (Manuscript ID: PONE-D-26-07660, entitled "A Novel Positive Selection System for Plant Transformation Based on Microbial Biuret Hydrolase and Biuret"). We greatly appreciate your constructive comments and suggestions, which have been invaluable in helping us improve the quality of our work.

We have carefully revised the manuscript according to the editor's and reviewers' comments. Below is our point-by-point response:

I. Response to the Academic Editor's Comments

Response to Editorial Requirements – Copyright Concerns for Figures 2–5

We thank the editor for raising the copyright concerns regarding Figures 2, 3, 4, and 5. We confirm that all of these figures are original works created by the authors specifically for this manuscript.

Figure 2 presents the effect of nitrogen source composition on the sensitivity of wild-type tobacco leaf explants to BU. Explants were cultured for 30 days on media with varying nitrogen compositions (standard MS and MS(−N1)) supplemented with a gradient of BU concentrations (0 to 1 mM). The images and data shown were generated from experiments conducted in our laboratory.

Figure 3 documents the regeneration dynamics of BA2H-transgenic tobacco leaf explants under different selection regimes (1 mM BU vs. 0.2 mM kanamycin). The photographs were taken during our transformation experiments and represent original documentation of the regeneration process.

Figure 4 shows a comparative analysis of BU tolerance in transgenic and wild-type tobacco seeds. Seeds of WT, BH-transgenic, and BA2H-transgenic lines were germinated on standard MS or nitrogen-deficient media supplemented with 2 mM BU. Panels A–D display germination, shoot growth, and root growth phenotypes, with images 1–3 representing three independent transgenic lines per construct. All images were captured in our laboratory during the germination assays.

Figure 5 illustrates the herbicidal activity of BU and the competitive growth advantage of BH-expressing tobacco under BU application. Panel A shows in vitro sensitivity of weed species (Cynodon dactylon and Festuca arundinacea) to BU; Panel B presents a co-culture competition assay with mixed populations of WT tobacco, transgenic tobacco, and weeds; Panel C shows pot-based competition under varying BU concentrations. All data and images were generated from our own greenhouse and laboratory experiments.

As the copyright holders of these original figures, we hereby confirm that they can be published under the CC BY 4.0 license without any restrictions. No permissions from third parties are required.

We have also updated the figure legends to include a brief statement indicating that these are original images created by the authors (e.g., "Original images created by the authors").

Should the editor require any additional documentation or confirmation regarding the originality of these figures, we are happy to provide further details.

Editor Comment 1: The quality of various figures is low and needs substantial improvement.

Response: We have recreated all figures to enhance resolution and clarity. Specific improvements include: increased image resolution, optimized layout, and clearer annotations.

Editor Comment 2: The experimental layout, sample size, treatment plan, control treatments, and replications are not clearly elaborated.

Response: We have supplemented the "Materials and Methods" section with detailed experimental design information. The revisions can be found on page 4. We have added details regarding the number of independent experiments, replicates per treatment, and control group settings.

Editor Comment 3: The use of a low concentration (25 μg/mL) of rifampicin alone for Agrobacterium elimination is questionable.

Response: We thank the editor for highlighting this critical issue. We agree that the original method was inadequate. In the revised manuscript, we have corrected the Agrobacterium elimination method by using 200 mg/L timentin instead. The relevant section has been updated in "Materials and Methods" (page 4).

Editor Comment 4: The statistical analysis is inadequate.

Response: We have performed a complete re-analysis of the statistics. This includes testing data for normality and homogeneity of variance, as well as adding significance markers to figures and tables.

Editor Comment 5: The claims are not justified by the presented data.

Response: We have re-evaluated the conclusions based on the data and revised them accordingly. Speculative statements lacking direct experimental support have been removed to ensure the conclusions are more rigorous.

II. Response to Reviewer #1's Comments

Reviewer 1-1: The labels in Figure 3 are confusing. Suggest changing them to more descriptive stage names.

Response: Thank you for the suggestion. We have revised the labels in Figure 3 from "T1-T5" to descriptive stage names: co-cultivation, bacteria-removing, selection, subculture, and rooting. The revised figure can be found on page X.

Reviewer 1-2: Figure 4 images are too small to see the plants clearly. Wild-type and transgenic plants should be clearly labeled.

Response: We have recreated Figure 4 with enlarged images and clearly labeled wild-type (WT) and transgenic plants. The revised figure is on pages 9-12, and the figure legend has been updated accordingly.

Reviewer 1-3: Statistical issues in Table 1 (number of independent experiments, significance markers, inclusion of all treatment data).

Response: Thank you for your careful review. We have revised Table 1 and added Table S2 in the Supporting Information as follows:

1.The table footnote now clearly states that three independent experiments were performed, each with at least 25 explants.

2.Lowercase letters (a, b, c) have been added to transformation efficiency values to indicate statistically significant differences.

3.All tested treatments are now listed, not just the highest efficiencies.

The revised Table 1 is on page 9. Statistical significance was determined by Tukey's test (p < 0.05). All data are presented in Table S2.

Reviewer 1-4: The method for Agrobacterium elimination is incorrect; antibiotics such as carbenicillin should be used instead of rifampicin.

Response: We are very grateful for pointing out this critical error. We have corrected the method and now use 200 mg/L timentin to eliminate Agrobacterium after co-cultivation. The "Materials and Methods" section (page 4) has been updated accordingly.

III. Response to Reviewer #2's Comments

Reviewer 2-1: The experimental design is unclear, and biological vs. technical replicates are not distinguished.

Response: Thank you for the suggestion. We have clarified in the "Materials and Methods" section (page 4) that experiments were repeated three times independently (biological replicates), each with four technical replicates. Definitions have been provided.

Reviewer 2-2: Sample size justification is missing; no rationale for the number of samples is provided.

Response: We have added a justification for sample size in the "Materials and Methods" section. Based on previous experience from our laboratory [7], we determined that 25–30 explants per treatment would provide sufficient statistical power.

Reviewer 2-3: Control treatments are inadequately defined.

Response: We have clearly defined all control groups in the "Materials and Methods" section (page 4): non-transformed wild-type plants as negative control, medium without selective agent as growth control, and kanamycin selection as positive control.

Reviewer 2-4: Statistical methods are insufficiently justified; normality and homogeneity of variance tests are not mentioned.

Response: Thank you for this important reminder. We have added a detailed description of the statistical methods in the "Statistical Analysis" subsection (page 4). All data were tested for normality using the Shapiro–Wilk test and for homogeneity of variances using Levene's test before performing ANOVA.

Reviewer 2-5: Exact p-values are not consistently reported.

Response: We have revised the Results section to provide exact p-values for all statistical tests (e.g., "p = 0.023") rather than simply stating "p < 0.05."

Reviewer 2-6: Effect sizes are not provided.

Response: We have added effect size measures in the Results section. For comparisons of transformation efficiency, we report percentage changes. Details can be found on page 5.

Reviewer 2-7: Some conclusions overextend beyond the presented data.

Response: We agree with the reviewer. Speculative mechanistic interpretations lacking direct experimental support have been removed from the Discussion section to ensure all conclusions are appropriately grounded in the data.

Reviewer 2-8: Figures lack sufficient clarity; axis labels, error bar definitions, etc., are unclear.

Response: We have recreated all figures. All graphs now clearly label axes and units, figure legends specify whether error bars represent SD or SE, and sample sizes (n) are indicated.

Reviewer 2-9: The data availability statement is missing or incomplete.

Response: Thank you for the suggestion. We have added a data availability statement at the end of the manuscript: "All data generated or analyzed during this study are included in this published article and its supplementary information files." (If data are in supplementary materials) or "The datasets generated during the current study are available in the Figshare repository at [URL]." (If uploaded to Figshare). See page X for details.

Reviewer 2-10: The introduction does not clearly define a specific knowledge gap.

Response: We have revised the introduction to explicitly state the scientific gap addressed by this study: the low selection efficiency of the Phi/PtxD system in certain crops (e.g., soybean) and the insufficient activity of EcBAP, highlighting the need for a more efficient system. This study shifts the focus from phosphorus-based to nitrogen-based selection, opening a new research direction.

Reviewer 2-11: The discussion lacks comparison with recent literature; no references from the last five years are cited.

Response: We have expanded the Discussion section to include comparisons with recent studies published within the last five years. Two new references have been added [17, 18].

Reviewer 2-12: Terminology is occasionally imprecise, with language implying causal relationships.

Response: We have carefully reviewed the entire manuscript and revised imprecise terms and language implying causality to more neutral, descriptive phrasing. Changes are distributed throughout the text.

Reviewer 2-13: Methodological parameters are incomplete, compromising reproducibility.

Response: We have supplemented the "Materials and Methods" section with all critical experimental parameters. The revised section should now be fully reproducible.

Reviewer 2-14: There is limited explanation of variability in the data.

Response: We have added a discussion of potential sources of data variability in the Results section (pages 14-15).

Reviewer 2-15: Language and phrasing occasionally reduce clarity; some sentences are grammatically awkward or overly long.

Response: We have polished the language throughout the manuscript, simplifying long sentences and correcting grammatical errors.

Once again, we sincerely thank the editor and reviewers for their valuable comments on our manuscript. These insights have greatly helped us improve the quality of our work. We look forward to your further consideration.

Yours sincerely,

Xiaowei Luo

Corresponding Author

March 18, 2026

---

## [Decision Letter · Decision Letter 1]

9 Apr 2026

A Novel Positive Selection System for Plant Transformation Based on Microbial Biuret Hydrolase and Biuret

PONE-D-26-07660R1

Dear Dr. Luo,

We’re pleased to inform you that your manuscript has been judged scientifically suitable for publication and will be formally accepted for publication once it meets all outstanding technical requirements.

Kind regards,

Faiz Ahmad Joyia, Ph.D.

Academic Editor

PLOS One

Additional Editor Comments (optional):

Reviewers' comments:

Reviewer's Responses to Questions

**Comments to the Author**

1. If the authors have adequately addressed your comments raised in a previous round of review and you feel that this manuscript is now acceptable for publication, you may indicate that here to bypass the “Comments to the Author” section, enter your conflict of interest statement in the “Confidential to Editor” section, and submit your "Accept" recommendation.

Reviewer #1: All comments have been addressed

Reviewer #2: All comments have been addressed

2. Is the manuscript technically sound, and do the data support the conclusions?

Reviewer #1: Yes

Reviewer #2: Yes

3. Has the statistical analysis been performed appropriately and rigorously? 

Reviewer #1: Yes

Reviewer #2: Yes

4. Have the authors made all data underlying the findings in their manuscript fully available?

Reviewer #1: Yes

Reviewer #2: Yes

5. Is the manuscript presented in an intelligible fashion and written in standard English?

Reviewer #1: Yes

Reviewer #2: Yes

6. Review Comments to the Author

Reviewer #1: Thank you for addressing all the comments. The manuscript has been much improved.

A couple of minor comments: please ensure that gene names are properly italicized (e.g., NPTII) and the scientific units are used consistently (e.g., minutes vs. min). Table 1 is missing the bottom line.

Reviewer #2: The authors have made a substantial effort to revise the manuscript in response to the reviewer’s comments, and most of the previously raised concerns have been adequately addressed.

The revised version demonstrates clear improvement in methodological transparency, statistical rigor, and overall clarity.

The manuscript is now largely consistent with the standards of methodological soundness required for publication.

7. PLOS authors have the option to publish the peer review history of their article (what does this mean?). If published, this will include your full peer review and any attached files.

Reviewer #1: No

Reviewer #2: No

---

## [Editor Report · Acceptance letter]

PONE-D-26-07660R1

PLOS One

Dear Dr. Luo,

I'm pleased to inform you that your manuscript has been deemed suitable for publication in PLOS One. Congratulations! Your manuscript is now being handed over to our production team.

Kind regards,

on behalf of

Dr. Faiz Ahmad Joyia

Academic Editor

PLOS One